# Locating Information in Large Language Models via Random Matrix Theory

## Abstract

As large language models (LLMs) become central to AI applications, gaining a deeper understanding of their inner workings is increasingly important. In this work, we analyze the weight matrices of pretrained transformer models – specifically BERT and Llama – using random matrix theory (RMT) as a zero-information hypothesis. While randomly initialized weights perfectly agree with RMT predictions, deviations emerge after training, allowing us to locate learned structures within the models. We identify layer-type specific behaviors that are consistent across all blocks and architectures considered. By pinpointing regions that deviate from RMT predictions, we highlight areas of feature learning and confirm this through comparisons with the activation covariance matrices of the corresponding layers. Our method provides a diagnostic tool for identifying relevant regions in transformer weights using only the trained matrices. Additionally, we address the ongoing debate regarding the significance of small singular values in the context of fine-tuning and alignment in LLMs. Our findings reveal that, after fine-tuning, small singular values play a crucial role in the models' capabilities, suggesting that removing them in an already aligned transformer can be detrimental, as it may compromise model alignment.

## 1 Introduction

Large language models (LLMs) have become foundational in deep learning, revolutionizing natural language processing tasks such as translation, text classification, and question answering (Vaswani et al., 2017; Yang et al., 2019; Touvron et al., 2023; Le Scao et al., 2023). Despite the well-documented success (Liu et al., 2019) of models like BERT (Devlin et al., 2018), the GPT series, and vision transformers (Dosovitskiy et al., 2021; Touvron et al., 2021; Liu et al., 2021), a thorough theoretical understanding of their inner workings remains elusive. Researchers have explored various facets of LLMs (Radford et al., 2019), yet key questions about how these models encode information and the roles of specific model components remain unanswered.

A potential avenue for deeper insights lies in the application of random matrix theory (RMT), which has been effective in neural networks for identifying structural properties and information density (Martin & Mahoney, 2021; Thamm et al., 2022; Staats et al., 2023). RMT has already shown promise in determining where information resides in models, particularly through analyzing the spectrum of weight matrices. As networks are initialized randomly, the weights precisely follow RMT predictions before training. After training, changes to the weights become visible when comparing them to RMT predictions. We build on these insights by leveraging RMT to pinpoint regions in LLMs where relevant features are learned, using deviations of the weight matrices from RMT predictions as indicators.

In this work, we study the weight matrices of pretrained BERT[1] and Llama-8B[2] models using RMT as a diagnostic tool. We find that certain types of matrices exhibit significant deviations from RMT predictions, while others remain close to their initialization. This pattern is consistent across different layers of the transformers and holds true for both the smaller BERT and the more powerful Llama-8B model. We identify the regions with the strongest deviations as areas of feature learning

---

[1] google-bert/bert-base-uncased
[2] meta-llama/Meta-Llama-3.1-8B

and confirm this through a comparison to the covariance matrix of the layer activations. Furthermore, we analyze the effect that the removal of groups of singular values and corresponding vectors from a fine-tuned BERT transformer has on the BoolQ validation accuracy. We find that the removal of groups in which the hypothesis of random vectors is less likely leads to significantly larger drops in validation accuracy. Our method allows us to pinpoint key areas in the transformer architecture using nothing more than the trained weight matrices.

Additionally, we contribute to the ongoing debate on the significance of small singular values, particularly in relation to fine-tuning and alignment in LLMs. Some studies suggest that small singular values are crucial for generalization (Hsu et al., 2022), while others argue that removing them can be beneficial (Sharma et al., 2023). We reconcile these views by showing that the importance of small singular values arises from the fine-tuning process conducted prior to reduction in Hsu et al. (2022). The findings of Perez et al. (2022) indicate that alignment can degrade LLM performance in certain tasks, which may explain the observed improvements when small singular values are removed. Our results suggest that reducing an already aligned transformer could be counterproductive, as it risks disrupting the model's alignment. All code to generate the figures is open source and available under Anonymous (2024).

## 2 RELATED WORK

RMT has been widely used as a calculational tool for performing statistical averages in the analysis of machine learning models. Early applications of RMT to neural networks, such as Pennington & Bahri (2017), analyzed the spectral properties of loss surfaces in deep learning, providing insights into learning dynamics. Building on this foundation, Baskerville et al. (2022) derived universal aspects of outliers in loss surfaces. Beyond its role in statistical analysis, RMT has been proposed as a tool for analyzing trained network weight matrices. Martin & Mahoney (2021) applied RMT to weight matrices by examining the learning dynamics of image recognition models through their spectra. Following up on this work, Martin et al. (2021) suggested that large outliers in the singular value spectrum are indicative of well-trained matrices. Further studies (Thamm et al., 2022; Levi & Oz, 2023) reinforced RMT's utility in understanding how networks evolve during training. They demonstrated that deviations from RMT predictions indicate where feature learning occurs, as opposed to *lazy learning* (Chizat et al., 2019), where weights remain close to their initial random state. These findings underscore RMT's potential for identifying regions of learned features without the need for training data.

Transformers present unique challenges in understanding information storage. Prior work by Jawahar et al. (2019); Reif et al. (2019) has shown that different layers specialize in storing distinct types of knowledge, while Aken et al. (2020) examined how semantic information is encoded in neuron activations. Hendel et al. (2023) explored how in-context learning in LLMs can be understood, suggesting that models implicitly create temporary task-specific vectors during inference. Tenney (2019) investigated where linguistic information is stored within BERT models, revealing that different layers capture various components of classical NLP tasks, such as syntax and semantics. Li et al. (2022) demonstrated that models can construct internal representations of environments – such as board game states – without explicit training, highlighting emergent capabilities. In Park et al. (2023), the question of whether binary concepts can be described by geometrical directions in the embedding space is investigated. Lee et al. (2024) identified directions within the network that encode toxicity, offering insights into how models can be aligned by subtracting harmful behavior patterns. Hernandez et al. (2023) examined how transformers encode relational knowledge, such as synonyms, suggesting that these relationships are captured through linear structures within the model's latent space.

Finally, the low-rank structure of features in neural networks has been explored. Yu & Wu (2023) highlighted that while transformer features often exhibit low rank, their weight matrices do not, revealing a complex relationship between representations and parameters. Positional encodings, crucial to transformer performance, have also been studied for their role in shaping the learned feature space (Tsai et al., 2019).

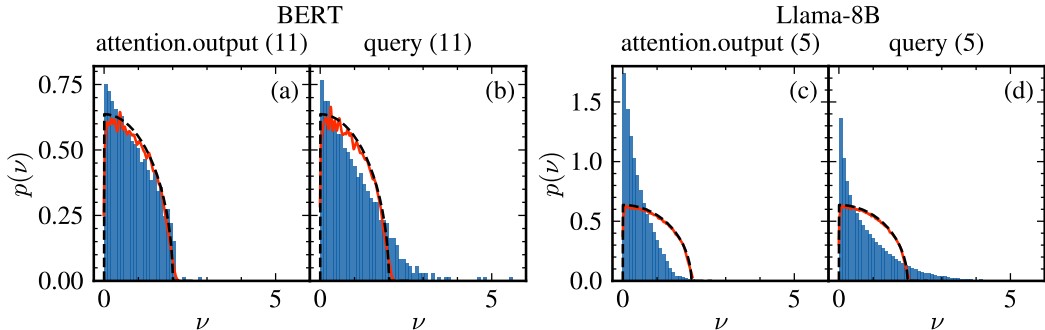

Figure 1: Singular value spectra of weight matrices from a pretrained BERT transformer ((a) and (b)) and a Llama-8B model ((c) and (d)), shown as blue histograms. For comparison, red curves represent the spectra of random matrices with identical dimensions and i.i.d. normally distributed entries with zero mean and standard deviation $1/\sqrt{m}$, mimicking freshly initialized network weights. The dashed black curves depict the MP distribution from Eq. 2. We observe that the empirical spectra deviate from the random control to varying degrees depending on the matrix type. Specifically, while the attention.output matrices exhibit only a few outliers and are dominated by regularization in the case of Llama, the query matrices display significant outliers for both the Llama-8B and BERT models.

## 3 SPECTRA OF LLMS WEIGHT MATRICES

To analyze the weight matrices of transformer networks, we perform a singular value decomposition (SVD) to decompose each weight matrix into its singular values and singular vectors. For a given weight matrix $W \in \mathbb{R}^{m \times n}$, the SVD factorizes $W$ into three matrices

$$W = USV^T \, , \tag{1}$$

where $U \in \mathbb{R}^{m \times m}$ and $V \in \mathbb{R}^{n \times n}$ are orthogonal matrices containing the left and right singular vectors of $W$, respectively, and $S \in \mathbb{R}^{m \times n}$ is a diagonal matrix containing the real, non-negative singular values of $W$.

In the limit of large matrix dimensions, $m, n \to \infty$, the distribution of singular values for matrices with independent and identically distributed (i.i.d.) random entries with finite variance is known to follow the Marchenko-Pastur (MP) law (Marčenko & Pastur, 1967)

$$P_{\mathrm{MP}}(\nu) = \begin{cases} \frac{n/m}{\pi \tilde{\sigma}^2 \nu} \sqrt{(\nu_{\max}^2 - \nu^2)(\nu^2 - \nu_{\min}^2)} & \nu \in [\nu_{\min}, \nu_{\max}] \\ 0 & \text{else} \end{cases} \tag{2}$$

$$\nu_{\substack{\max \\ \min}} = \tilde{\sigma}(1 \pm \sqrt{m/n}) \, , \quad \tilde{\sigma} = \sigma\sqrt{n} \, . \tag{3}$$

In the context of weight matrices in neural networks, although the dimensions are finite, they are often large enough for the Marchenko-Pastur distribution to approximate the singular value spectrum of randomly initialized weights well. After training, we can compare the empirical spectrum to the MP distribution to assess deviations resulting from the optimization process. Typically, the bulk of singular values remains close to the MP distribution, while significant deviations may indicate learned features. We illustrate this in Fig. 1, where the dashed black lines represent the MP law from Eq. 2, and the red curves show the broadened spectra of random square matrices with variance $1/m$ of the matrix elements. The figure displays the spectra of the query and attention.output matrices from the eleventh block of a pretrained BERT transformer (left panels) and from the fifth block of a pretrained Llama-8B model (right panels).

During training, certain directions in the weight matrices become more significant, leading to outliers in the singular value spectrum (Staats et al., 2023). It has been suggested that large outliers in the spectrum are indicative of well-trained matrices (Martin et al., 2021). This is in line with previous work (Thamm et al., 2022), which found that models trained in the lazy regime retain spectra identical to the MP distribution and generally perform worse than models trained in the rich or feature learning regime, where the spectra exhibit significant changes.

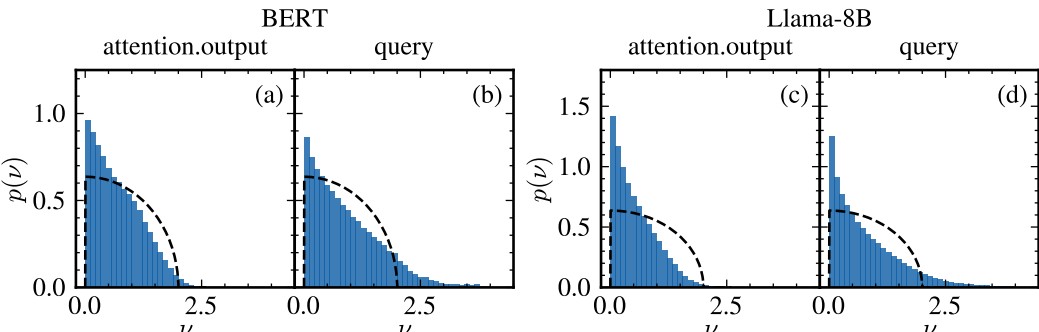

Figure 2: Averaged singular value spectra of the query and attention.output matrices across all layers of a pretrained BERT transformer ((a) and (b)) and a Llama-8B model ((c) and (d)), shown as blue histograms. The dashed black curves represent the MP distribution for reference. We find that the query matrices exhibit significantly more outliers than the attention.output matrices in both models. These observations suggest that stronger feature learning occurs in the query matrices compared to the attention.output matrices.

In Fig. 1, we observe that for both the pretrained Llama-8B and BERT transformers, the attention.output matrices have significantly fewer outliers in the singular value spectrum compared to the query matrices. We interpret this behavior as an indication that feature learning predominantly occurs in the query matrices, where the weights undergo substantial changes, while the attention.output matrices remain closer to their initial random state, reflecting lazy learning.

Moreover, the spectra of the Llama-8B model show stronger deviations from the initial distribution than those of the BERT model. We interpret this, in line with findings in vision models (Martin et al., 2021), as evidence of more effective learning in the Llama-8B model. When averaging the spectra over all matrices of the same type across all layers of the transformers, this effect persists, as shown in Fig. 2. We find that the attention.output matrices rarely produce outliers above 2.5, a common singular value for query matrices in these models. We later verify that the singular values and corresponding vectors outside the Marchenko-Pastur region indeed correspond to learned features by studying their overlap with the activation covariance matrix.

## 4 SINGULAR VECTORS OF WEIGHT MATRICES

In random matrices with i.i.d. entries of finite variance, the entries of a singular vector $v$ of length $n$ are expected to follow a normal distribution with a standard deviation of $1/\sqrt{n}$

$$P(v_i) = \frac{1}{\sqrt{2\pi/n}} \exp\left(-\frac{1}{2}v_i^2 n\right) \ . \tag{4}$$

To identify deviations from this expected behavior in the weight matrices of transformer networks, we perform Kolmogorov-Smirnov (KS) tests on the singular vectors. Specifically, we conduct Monte Carlo sampling of normalized Gaussian vectors to generate synthetic data and compare their empirical cumulative distribution functions (CDFs), denoted as $C_{\text{emp}}^{(k)}$, to the theoretical Gaussian CDF $C_{\text{G}}(x) = \frac{1}{2} + \frac{1}{2}\text{erf}\left(\sqrt{n/2}\,x\right)$. The KS statistic for each sampled vector is calculated as the supremum of the absolute difference between the empirical and theoretical CDFs

$$D^{(k)} = \sup_x \left| C_{\text{emp}}^{(k)}(x) - C_{\text{G}}(x) \right| \ . \tag{5}$$

By sampling many such vectors, we obtain the distribution of expected deviations $D_c$ for perfectly random data. For each singular vector $v$ from the weight matrices, we compute its KS statistic $D^{(v)}$ and determine the corresponding $p$-value using the cumulative distribution function $C_{D_c}$ of $D_c$ via $p = 1 - C_{D_c}(D^{(v)})$. Under the null hypothesis that the singular vector entries are normally distributed, the $p$-values are uniformly distributed in $[0,1]$. We note that due to the normalization

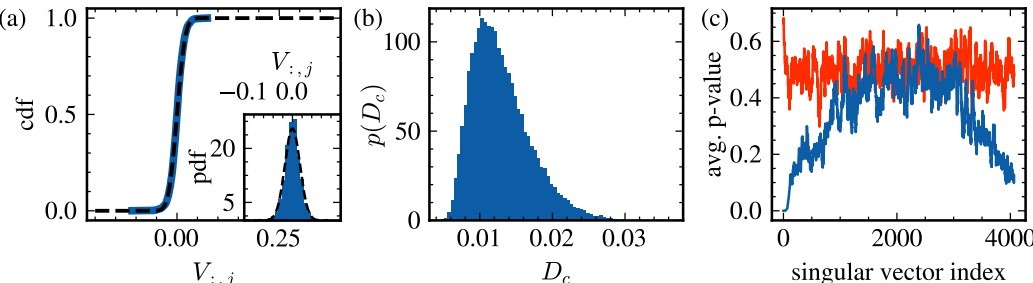

Figure 3: Analysis of the singular vectors of the attention.output matrix from block 20 of the pretrained Llama-8B model. (a) The cumulative distribution function (cdf) of the entries of a specific singular vector (blue line) compared to the theoretical Gaussian cdf (black dashed line). The inset shows the probability density function (pdf) of the entries. (b) The distribution of the Kolmogorov-Smirnov (KS) statistic $D_c$ obtained from synthetic random Gaussian vectors, used to compute $p$-values for the empirical singular vectors. (c) Averaged $p$-values for the singular vectors (blue line), compared to a random control (red line). We observe that the singular vectors corresponding to the largest and smallest singular values deviate significantly from randomness.

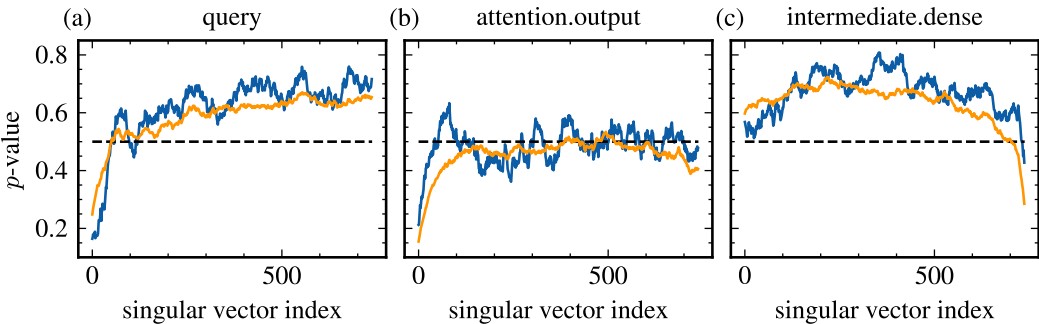

Figure 4: Averaged $p$-values from KS statistics comparing the entries of singular vectors to the normal distribution for selected weight matrices in a pretrained BERT transformer. Blue lines represent the $p$-values for weight matrices from the fourth transformer block, while orange lines represent the average $p$-values for the respective type of weight matrix across all transformer blocks. The dashed horizontal line indicates the average $p$-value for the random control. Lower $p$-values suggest deviations from the initial random weight matrix, which we interpret as evidence of learned information during pretraining.

constraint of singular vectors, which introduces correlations between their entries, standard KS test tables are not applicable. Therefore, we compute custom test statistics using the Monte Carlo approach described above.

Figure 3 (a) illustrates the probability density function (pdf) and cumulative distribution function (cdf) of a right singular vector from a pretrained Llama-8B attention.output matrix. The expected distributions for synthetic Gaussian data are depicted in panel (b). To identify meaningful deviations from randomness across thousands of singular vectors, we define a local average of the $p$-values

$$p_{\text{avg}}(\boldsymbol{v}_j) = \frac{1}{15} \sum_{i=j-7}^{j+7} p(\boldsymbol{v}_i) \,. \tag{6}$$

This averaging smooths out fluctuations and, for uniformly distributed $p$-values, results approximately in a Gaussian distribution with a mean of 0.5 and a standard deviation of 0.05. Panel (c) in Figure 3 shows these averaged $p$-values for the right singular vectors of a Llama-8B attention.output matrix. We observe significant deviations from the RMT prediction in the singular vectors associated with the largest and smallest singular values.

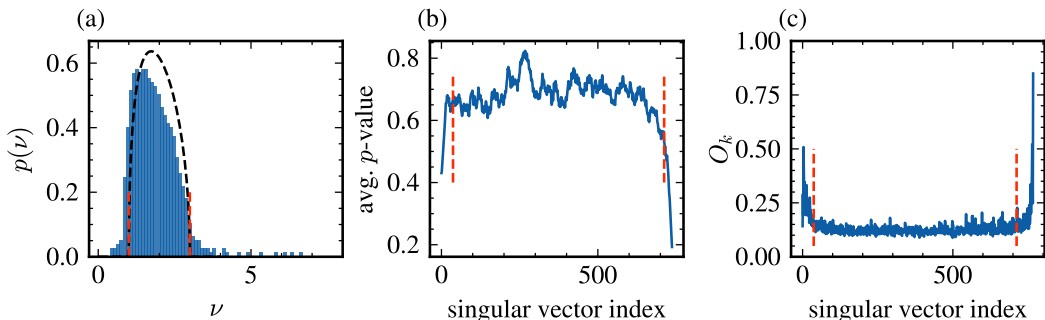

Figure 5: RMT analysis of the intermediate.dense matrix from the first block of a BERT transformer. (a) The empirical singular value spectrum (blue histogram) shows clear outliers on both the left and right sides relative to the MP distribution (black dashed line). Left-side outliers are possible due to the aspect ratio differing from one. (b) The $p$-values of the singular vectors are reduced in both these regions, indicating deviations from randomness. (c) By computing the activation covariance matrix from activations entering this layer (using the BoolQ training dataset) and calculating the maximal overlap of its eigenvectors with the singular vectors, we find that the regions outside the MP curve (indicated by dashed red lines) have a large overlap with the eigenvectors of the activation covariance matrix. In contrast, vectors inside the MP spectrum do not. We interpret the regions that deviate from RMT predictions as corresponding to learned features.

Figure 4 presents the averaged $p$-values for the right singular vectors of a pretrained BERT model. The blue curves represent a single matrix from the fourth block, while the orange curves represent averages over all blocks. We find that the singular vectors corresponding to the largest singular values deviate significantly from randomness for both the query matrix (panel a) and the attention.output matrix (panel b). This holds true for individual matrices as well as the averages, supporting the notion of matrix-type-specific learning. In contrast, for the intermediate.dense matrix, significant deviations occur in the singular vectors corresponding to the smallest singular values. Later, we demonstrate that these singular vectors have a strong overlap with eigenvectors of the activation covariance matrix, indicating their importance in feature representation.

It is worth noting that regions where the averaged $p$-values are significantly above 0.5 are due to the orthogonality constraints of the singular vectors (Staats et al., 2023). When some vectors have a significant mean, the orthogonality condition forces the other vectors to adjust to maintain zero mean overall, introducing correlations between their entries.

## 5 ACTIVATION COVARIANCE MATRIX

In the following, we investigate whether the non-random regions in the weight matrices correspond to features learned by the transformer. This is accomplished by comparing the activation covariance matrix, computed from the activations entering a layer, to the weight matrix of that same layer. Formally, we compute the activation covariance matrix $F^{(\ell)}$ for layer $\ell$ by averaging over $n_{\text{ex}}$ input examples, indexed by $i_{\text{ex}}$, and $n_{\text{t}}$ tokens, indexed by $j_{\text{t}}$. Let $\boldsymbol{x}^{(\ell)}_{i_{\text{ex}},j_{\text{t}}}$ denote the activations entering layer $\ell$. The activation covariance matrix is then given by

$$F^{(\ell)}_{nm} = \frac{1}{n_{\text{ex}}n_{\text{t}}} \sum_{i_{\text{ex}},j_{\text{t}}} x^{(\ell)}_{i_{\text{ex}},j_{\text{t}},n} x^{(\ell)}_{i_{\text{ex}},j_{\text{t}},m} \ . \tag{7}$$

This matrix is symmetric and therefore has an orthonormal eigenvector basis. We denote the eigenvectors by $\boldsymbol{f}^{(\ell)}_i$ and the corresponding eigenvalues by $\lambda^{(\ell)}_i$.

To compare these eigenvectors with the weight matrix, we consider how the activations $\boldsymbol{x}^{(\ell)}$ enter layer $\ell$. Specifically, we have $W^{(\ell)}\boldsymbol{x}^{(\ell)} + \boldsymbol{b}^{(\ell)} = USV^T\boldsymbol{x} + \boldsymbol{b}$, where $U, S, V$ are from the singular value decomposition of $W^{(\ell)}$. This equation shows that the neuron activations are directly mapped onto the basis of the right singular vectors $V$. We then ask whether a specific eigenvector of the

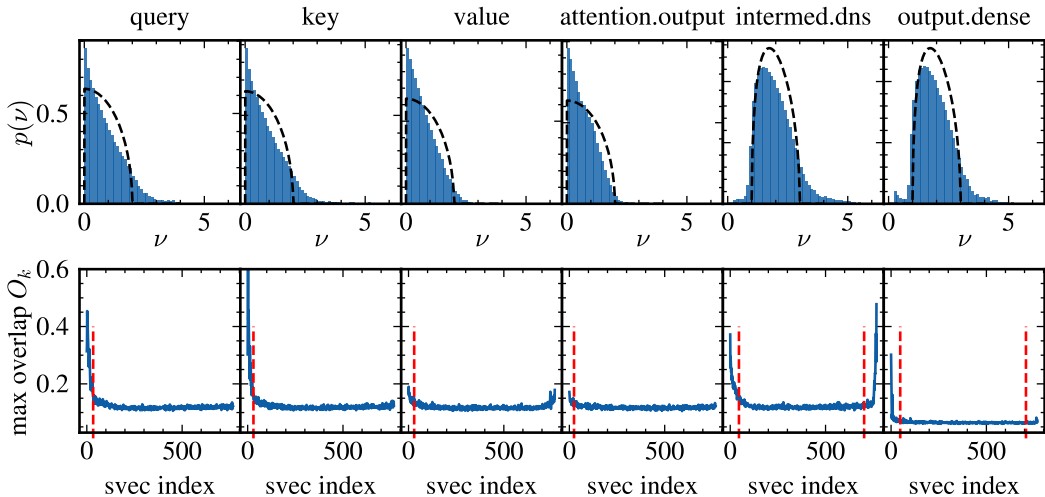

Figure 6: Comparison between the average singular value spectra (upper panels) and the averaged maximal overlap of singular vectors with the eigenvectors of the activation covariance matrix (lower panels) for different weight matrices in a pretrained BERT model. The activation covariance matrix is computed on the BoolQ training set. The attention.output and value matrices have spectra that remain close to the initial random distribution and show limited overlap with eigenvectors of the activation covariance matrix. In contrast, the key and query matrices display larger deviations from the initial distribution, including significant outliers, and show substantial overlap with eigenvectors of the activation covariance matrix. The red dashed lines indicate the boundaries of the MP distribution. The areas with significant overlap correspond well to regions outside of this distribution. These findings suggest that feature learning occurs predominantly in the key, query, and intermediate.dense matrices, but not in the attention.output and value matrices.

activation covariance matrix corresponds to one of the right singular vectors of the weight matrix by computing

$$O_k^{(\ell)} = \max_j(\boldsymbol{v}_k^{(\ell)} \cdot \boldsymbol{f}_j^{(\ell)}), \quad j \in \{1, 2, ..., n\} . \tag{8}$$

This measure quantifies the extent to which the singular vectors capture specific features of the activation covariance matrix, and hence the data. In our analysis, we consider the activation covariance matrix computed from the BoolQ training dataset using a pretrained BERT transformer.

Figure 5 illustrates the agreement between RMT results and our analysis of the activation covariance matrix for the intermediate.dense matrix of a BERT transformer. The singular value spectrum exhibits both left and right outliers (left panel), and the $p$-values of the corresponding right singular vectors are reduced for both the largest and smallest singular values. Notably, these regions coincide with where the singular vectors have a large overlap with the activation covariance matrix (right panel). We find overlap values above $0.5$ for the singular vectors corresponding to the smallest and largest singular values, which is a significant overlap in a 768-dimensional space. The region between the two dashed lines represents the Marchenko-Pastur prediction computed with a standard deviation of $1/\sqrt{m}$. Within this region, the overlap with the activation covariance matrix is significantly smaller.

To demonstrate that these findings are general, we compute the activation covariance matrix for each layer, determine the maximal overlaps for each matrix, and then average these maximal overlaps. The results are shown in Figure 6 (lower panel), along with the corresponding averaged spectra (upper panel). We make the following observations: First, the intermediate.dense matrix exhibits a strong overlap in singular vectors with high indices (i.e., small singular values), which aligns well with the $p$-values of these matrices shown in Figure 4, where the $p$-values drop significantly for smaller singular values. Second, for both the query and key matrices, there are pronounced outliers in the spectrum (extending beyond a value of 3) that correspond to singular vectors with large overlaps with the activation covariance matrix. This is not the case for the value and attention.output

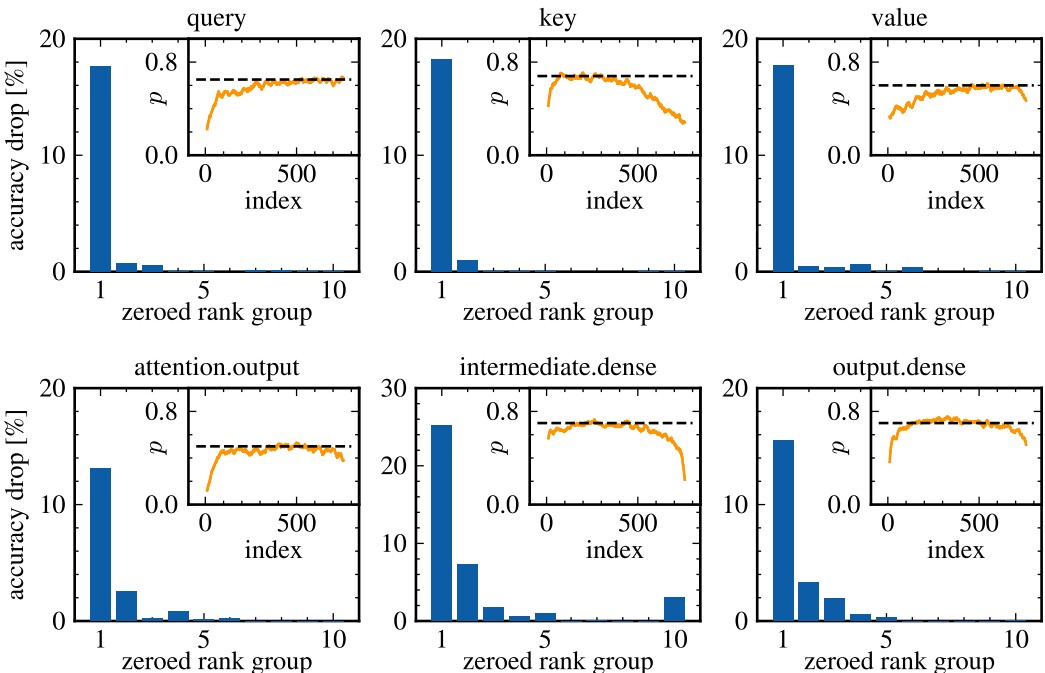

Figure 7: Impact on validation accuracy when removing blocks of singular values from all weight matrices of a given type in a fine-tuned BERT transformer evaluated on the SuperGLUE-BoolQ dataset. Each block contains $10\%$ of the singular values; block 1 corresponds to the largest singular values, and block 10 to the smallest. The main plots show the decrease in validation accuracy after setting these singular values to zero. The insets display the average $p$-values of the singular vectors for each matrix type, averaged over all layers, with the horizontal dashed line indicating the plateau value as a guide to the eye. Values below this plateau suggest learned information during pretraining. All results are averaged over five fine-tuning runs with different random seeds for initializing the transformer heads. Removing the largest singular values leads to the greatest accuracy drops across all matrix types, which is expected since significant alterations to the weight matrices affect the downstream signal most. Strong deviations from RMT predictions in the corresponding singular vectors are observed for all matrices except the intermediate.dense matrices. Interestingly, for the intermediate.dense matrices, the singular vectors corresponding to the smallest singular values exhibit reduced $p$-values, indicating learned information, as confirmed by the accuracy drop when these singular values are removed. However, low $p$-values do not always correspond to a performance drop, as the learned information during pretraining may not be utilized in a given downstream task, as seen with the key matrices.

matrices, where the largest outliers remain below 2.5. This effect, combined with the observation that the attention.output matrix remains very close to the original MP shape and shows very little overlap with the activation covariance matrix, strongly indicates that these matrices are not trained in the feature learning regime.

# 6 REMOVING SINGULAR VALUES

In the previous sections, we demonstrated a significant overlap between the singular vectors of weight matrices – specifically in regions where these matrices deviate from RMT predictions – and the eigenvectors of the activation covariance matrix. To further assess the relevance of these singular values and their corresponding singular vectors, we conducted experiments where we removed specific groups of singular values. Removing a singular value $\nu_r$ from a weight matrix is achieved

by setting it to zero in $S$ and reconstructing the weight using the original singular vectors

$$W = USV^T, \quad \longrightarrow \quad \tilde{S}_{ii} = \begin{cases} \nu_i & \text{for } i \neq r \\ 0 & \text{else} \end{cases} \quad \longrightarrow \quad \tilde{W} = U\tilde{S}V^T \ . \tag{9}$$

Because removing a single singular value in a full transformer model has negligible effect, we grouped the singular values of each matrix into ten equally sized sets and removed these sets individually from the transformer. To assess the effect of removing singular values, we fine-tuned a pretrained BERT transformer using five different random seeds for initializing the model heads on the BoolQ dataset, achieving an average validation accuracy of 73.6%. We then removed one of the singular value deciles from a specific matrix type in all layers; for example, we set the largest 10% of singular values in each query matrix to zero and measured how the validation accuracy dropped compared to the full model.

We present the results in Fig. 7, which shows good agreement between the regions that deviate from RMT and the regions that are crucial for the transformer's test performance. As expected, for all matrix types, the removal of the largest singular values leads to the greatest accuracy drops. This is corroborated by the $p$-values of the right singular vectors; in five out of the six cases, we observe significant drops in $p$-values for vectors corresponding to the largest singular values. As a reference for the $p$-value drops, we consider the plateau value, indicated by the dashed black line as a visual guide. In the case of the intermediate.dense matrices, the singular vectors corresponding to small singular values have the largest deviations from RMT. This is reflected in a large accuracy drop when removing these small singular values. Although less pronounced than in the intermediate.dense matrices, the key matrices also exhibit significant RMT deviations for singular vectors corresponding to small singular values. However, when we tested the impact of removing these small singular values from the key matrices on the BoolQ dataset, we did not observe a significant effect on the generalization performance. Such behavior is expected when the information learned during pretraining is not utilized by the downstream task (see Appendix C for an example on the SuperGLUE-WiC dataset, where removing these small singular values impacts performance).

Although one might consider using this scheme to reduce the network size, we find that removing larger portions of the "random" parts of the spectrum significantly degrades the network's performance. To understand this behavior, we consider the case where a weight matrix in the network architecture is completely random and is kept frozen during training. In this scenario, the network is still able to learn, but the removal of small and intermediate singular values from the random weights significantly impacts the overall performance, as the subsequent layers are sensitive to small changes in the random matrix. In Appendix A, we demonstrate that matrices which have learned robust features are highly resilient to such removal, whereas removing singular values from a random matrix destroys the subtle details that subsequent layers depend upon.

## 7  FINE-TUNING

Recent studies have debated the relevance of small singular values in transformer networks. Some argue that these values are crucial for network performance (Hsu et al., 2022), while others have observed performance improvements when they are removed (Sharma et al., 2023). Our RMT analysis reveals significant deviations only for some of the smaller singular values and their corresponding vectors, providing a diagnostic tool to assess their importance. This finding supports the notion that small singular values can play a significant role.

In Figure 8, we investigate the relevance of singular values before and after fine-tuning by removing deciles of singular values from all weight matrices simultaneously. We observe a clear difference between the two scenarios: when singular values are removed before fine-tuning and the model is fine-tuned afterward, the performance is not significantly affected by the removal. However, when the model is fine-tuned first and singular values are removed afterward, the performance drops significantly, indicating that these singular values are crucial to the model's performance after fine-tuning. This observation explains the differences found in the literature. Hsu et al. (2022) fine-tuned first and found that the small singular values are important to the network, while Sharma et al. (2023) found it beneficial to remove them from a model that is directly evaluated on a benchmark without fine-tuning. We interpret this behavior as evidence that fine-tuning, and potentially alignment, are encoded in the smaller singular values and their corresponding vectors. Notably, aligning LLMs

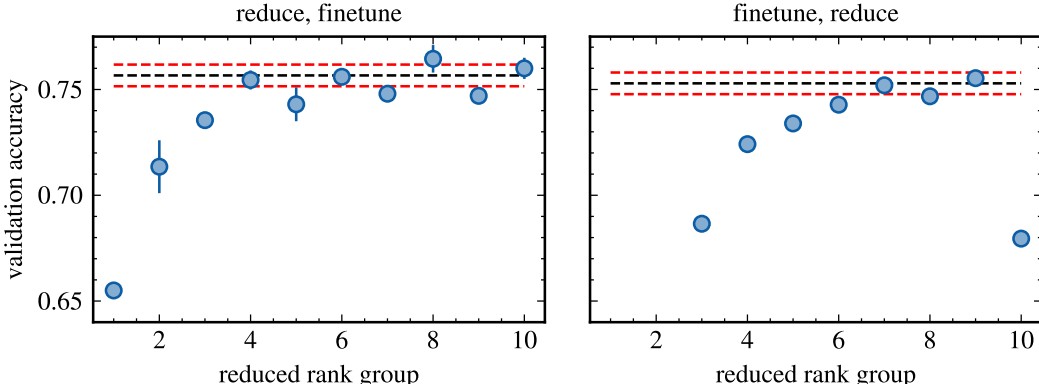

Figure 8: Effect on validation accuracy when removing deciles of singular values from all matrices except the embedding weights of a BERT transformer. Decile one corresponds to the largest 10% of the singular values, and its removal results in large accuracy drops. This is observed both when removing singular values and then fine-tuning the model (left panel), and when fine-tuning the model first and then removing singular values (right panel). For the smallest singular values (block 10), the scenario changes markedly. When reducing first, removing small singular values has negligible impact, with changes in final validation accuracy within the error bars of five different full-model seeds. However, when fine-tuning first, removing the smallest singular values leads to significant accuracy drops. This demonstrates that fine-tuning primarily affects the smallest singular values and their corresponding vectors.

can sometimes degrade performance on reasoning tasks (Perez et al., 2022), which may explain the improvements observed by Sharma et al. (2023) when small singular values are removed. We conclude that small singular values may be crucial for the alignment of LLMs, and we speculate that reducing an already aligned model by removing these singular values could be detrimental, as it may eliminate the alignment.

## 8 CONCLUSION

In this paper, we used random matrix theory (RMT) to analyze the weight matrices of BERT and Llama-8B models. Our findings show that certain weight matrices exhibit significant deviations from RMT predictions, indicating areas where active feature learning occurs. In contrast, other weight matrices, such as the attention.output matrix, remain close to their initial random state, suggesting that limited feature learning takes place. These deviations from RMT are consistent across all layers and persist when moving from BERT to Llama-8B, highlighting a potential structural pattern in transformer architectures.

We supported our hypothesis that deviations from RMT predictions correspond to learned features through an analysis of the activation covariance matrices of a BERT transformer. We identified a strong overlap between the weight's singular vectors in regions that deviate from RMT predictions and the eigenvectors of the corresponding covariance matrix of activations entering the layer. Furthermore, we found that removing regions of the weight matrices that deviate most from RMT predictions leads to significant performance drops, emphasizing the importance of these regions.

Additionally, we provided clarity on the ongoing debate regarding the importance of small singular values in LLMs. Our results show that while small singular values may not be crucial during pre-training, they become highly relevant during the fine-tuning process. Removing these small singular values after fine-tuning leads to significant accuracy drops, suggesting that fine-tuning refines the model primarily through small singular values and their corresponding vectors.

Overall, our work provides a diagnostic tool for identifying critical regions in transformer models based solely on their weight matrices and offers a new perspective on the role of singular values in model fine-tuning and alignment. These findings can inform future efforts to optimize transformer architectures and help explainable AI researchers pinpoint regions of particular interest.

REPRODUCIBILITY STATEMENT

To ensure reproducibility, we have uploaded all necessary code and materials to generate the figures in a Zenodo archive (Anonymous, 2024). The provided folders offer different entry points depending on the user's requirements: (i) We include Jupyter notebooks that load pre-saved data to quickly reproduce the figures. (ii) For plots that are less resource-intensive, we provide notebooks that directly generate the data. (iii) For resource-intensive tasks, we provide SLURM scripts that automate job submissions, along with the full set of hyperparameters used for fine-tuning. This structure ensures ease of access for quick figure generation while also supplying full details for in-depth replication of our experiments.

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

# A    LAZY LEARNING

It is possible to train neural networks in the lazy regime where the final weights of the trained model are very close to the initial ones (Chizat et al., 2019). By rescaling the input of the softmax function in the final layer by a constant $\alpha > 1$

$$a_L = \text{softmax}\left(\alpha(\mathsf{W}_L \boldsymbol{a}_{L-1} + \boldsymbol{b}_L)\right) ,$$

we achieve that very small changes in the output logits prior to the softmax function have a large effect on the output after the softmax function. To allow for learning with a usual learning rate, the loss is changed to

$$l(\boldsymbol{W}, \boldsymbol{b}) = -\frac{1}{N\alpha^2} \sum_{k=1}^{N} \boldsymbol{y}^{(k)} \cdot \ln(\boldsymbol{a}_{\text{out}}^{(k)}) , \qquad (10)$$

to incorporate the large differences in the output activations $a_L$ induces by small weight changes.

To investigate the effect that the removal of lazy parts from a neural network has on the test-accuracy, we train a fully connected network with layer dimensions [3072, 512, 512, 512, 10] on the Cifar-10 dataset, both in the lazy regime ($\alpha = 15$) and with the usual softmax activation function ($\alpha = 1$). The model trained with ($\alpha = 1$) reaches $53\%$ test-accuracy while the one trained in the lazy regime achieves $44\%$.

We now analyze the normalized test-accuracy drop when removing singular values in the three layers with $512$ singular values in Fig. 9. We observe that the removal of the smallest $20\%$ of the singular values has a negligible effect on the test accuracy of the network with $\alpha = 1$, while the accuracy of the lazy network drops significantly. This is the case despite the $\alpha = 1$ network having a much higher starting accuracy. The curves remain separated up to the point where both approach random guessing, for $90\%$ of the singular values removed. This indicates that the removal of a seemingly random area of the network might still negatively impact the generalization performance.

To further test this hypothesis, we train two models with layer sizes [3072,512,512,256,256,10], where for the second model, we freeze the first two layers during training. The full model achieves $53.5\%$ while the frozen model reaches $46\%$. When removing singular values in the first two layers we account for the magnitude of the singular values by setting $80\%$ of the singular value mass $M = \sum_i \nu_i$ to zero. We find that the model with frozen layers goes to random guessing, while the network with trained layers remains at $50\%$ test accuracy. This demonstrates that layers after a lazy trained matrix, can depend strongly on small details of the signal. Such details can easily be destroyed in a potential network compression algorithm, which results in bad performance. On the other hand, strong feature learning seems to result in a more robust network.

# B    LLAMA SPECTRA

To complete the picture of specific spectra shown in the main manuscript, we show the averaged spectra of all matrix types present in the pretrained Llama-8B model in Fig. 10. We observe that in

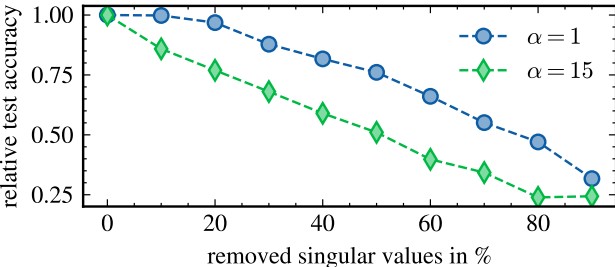

Figure 9: Removing singular values from all layers in a multi-layer perceptron trained in the feature learning regime (blue curve) and in the lazy regime (green curve). We find that the removal from the lazy regime is very difficult without losing accuracy.

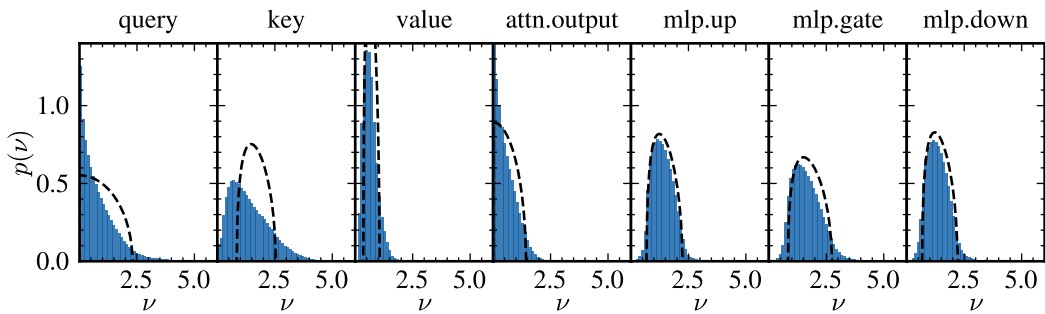

Figure 10: Average Spectra for each matrix type for Llama-8B model. We see that similar to BERT, query and key matrices have pronounced outliers while the attention.output and value matrices do not.

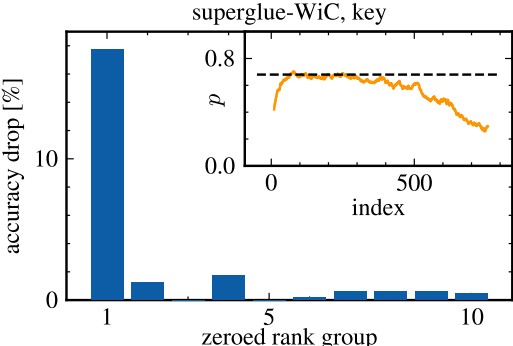

Figure 11: Example for removal from a key matrix, similar to main text Fig. 7, where the information stored in the smallest singular values is accessed by the superglue-WiC task.

general, regularization appears to be much stronger than in the BERT models, significantly shifting some of the spectra towards small singular values. We incorporate this in the MP theory by using the empirical variance of the matrix instead of $1/m$. In particular, if very little learning occurs (i.e. gradient updates are small implying $\alpha\partial_W\mathcal{L} \simeq 0$), the $L_2$ regularisation keeps the Marchenko-Pastur distribution intact as the learning dynamics

$$\partial_t W = -\alpha\partial_W\mathcal{L} - \lambda W \simeq -\lambda W \implies W(t) = \exp(-\lambda t)W_0 \tag{11}$$

only rescales the matrix. Here, $\alpha$ is the learning rate, $\lambda$ is the strength of the $L_2$ regularization, and $W_0$ is the initial weight matrix following the MP law.

Nevertheless, the key observations described for the BERT model in the main text hold true. The value and attention.output matrices create very little outliers and their singular values spectra remain below $\nu_i = 2.5$. In contrast, the corresponding matrices with identical shapes (key and query, respectively) have significantly larger values and more pronounced outliers. This further supports our hypothesis of lazy learning in the Value and attention.output matrix.

## C  EXAMPLE FOR SIGNIFICANCE OF SMALL SINGULAR VALUES IN KEY MATRICES OF BERT

We showed in the main text how the removal of singular values that correspond to singular vectors that deviate from the RMT prediction leads to a particularly large accuracy drop. While this was generally the case, the key matrices of BERT showed a significant deviation from RMT in their singular vectors corresponding to smaller singular values, not reflected in an accuracy drop on BoolQ when removing them. We argued that the information learned in these singular vectors during pre-

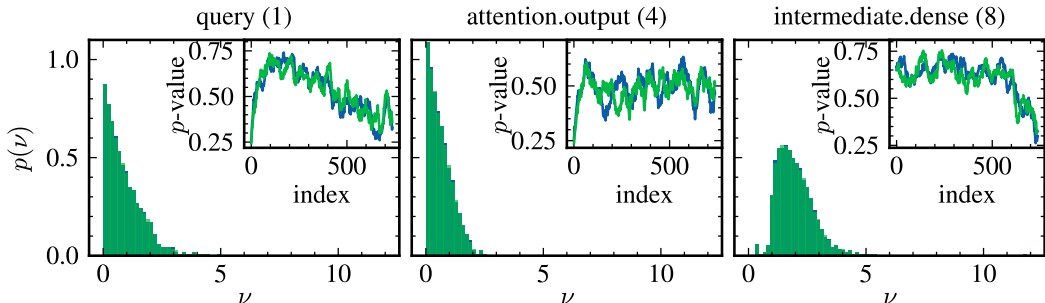

Figure 12: Effects on the spectra and $p$-values for specific matrices when fine-tuning a BERT trans-former on the superglue-BoolQ dataset. For all three matrices, we observe little to no changes in the spectra (pretrained: blue, fine-tuned: green). However, the $p$-values of the singular vectors with small or intermediate singular values do change slightly, indicating that fine-tuning may take place in directions other than the ones corresponding to the largest singular values.

training is not accessed by the BoolQ dataset and show an example where the small singular values do play a role for the superglue-WiC dataset in Fig. 11.

## D    FINE-TUNING WEIGHTS

We concur with earlier findings, which suggest that fine-tuning on datasets like glue (Wang et al., 2019b) and superglue (Wang et al., 2019a) induces minimal changes in the model's weights and does not substantially impact the spectrum of the model. This is supported by our RMT analysis in Fig. 12, where the spectra remain nearly unchanged after fine-tuning. Similarly the $p$-values of the corresponding singular vectors change very little. This is especially pronounced for the larger singular vectors indicating that fine-tuning may be happening particularly in directions that differ from the largest singular vectors.

## E    ACTIVATION COVARIANCE MATRIX

In the main text, we compute several activation covariance matrices and analyze the overlap of their eigenvectors with the singular vectors of the weight matrices. However, the activation covariance matrix is an interesting object to study on its own. For completeness, we provide spectra of the activation covariance matrix in Fig. 13. We show activation covariance matrices of a pretrained BERT model, where the activations are obtained using the BoolQ training dataset. We compute the

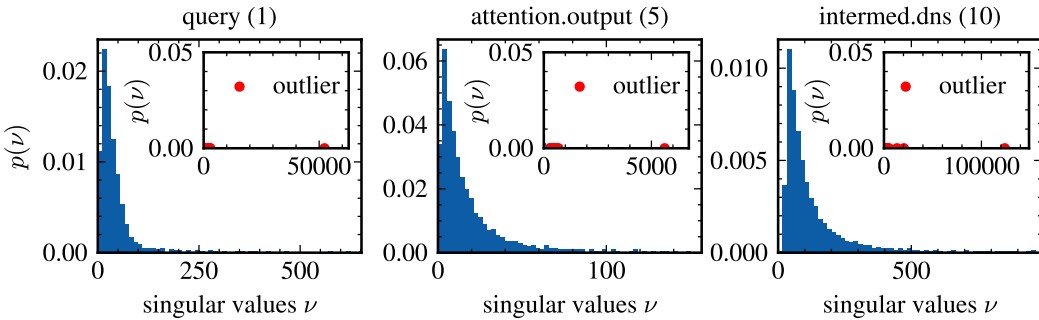

Figure 13: Activation covariance matrix of several matrices computed for the BoolQ training dataset. We find very large outliers which are most likely due to the positional encoding in BERT. We again find that that outliers in the attention output matrix are much smaller than the ones of other matrices.

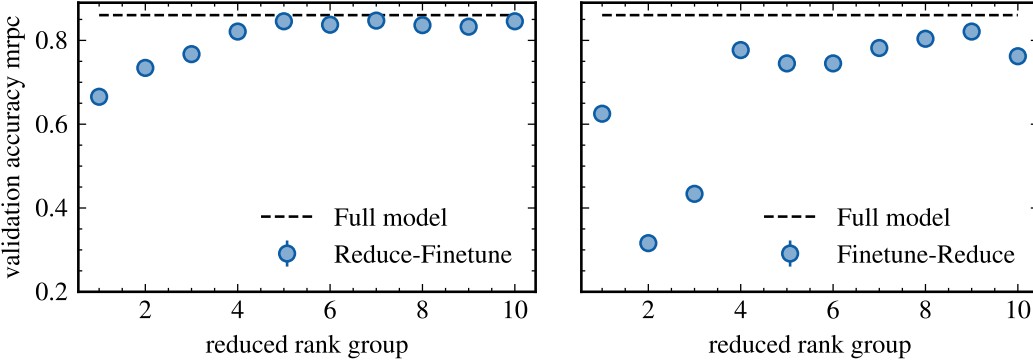

Figure 14: Reduction of a BERT transformer based on a projection onto the eigenvectors of the corresponding activation covariance matrix. We find that similar to the singular values and vectors, the eigenvectors corresponding to the smaller eigenvalues are not important when reducing first (left panel). However, when reducing after fine-tuning (right panel), these eigenvectors are suddenly more important than some of the larger percentiles.

activation prior to the considered matrices. The displayed activation covariance matrices have large outliers, which are completely beyond the rest of the spectrum. These outliers are most likely due to the positional encoding which has a significant influence on the activation covariance matrix as it occurs in every batch. We see that the pattern of much smaller outliers in the attention output matrix also repeats for the activation covariance matrix.

# F   ACTIVATION COVARIANCE MATRIX PROJECTION

We showed in the main text that the removal of the smallest singular value percentiles leads to a significant reduction in validation accuracy. To show that other methods that reduce the rank of a matrix also fall into the trap of removing important information encoded in the smallest singular values, we also showcase a different method. We apply a projection onto the eigenvectors $\boldsymbol{f}$ of the activation covariance matrix $F$ as described in Ashkboos et al. (2024). We therefore convert the layer-norm to RMS-norm and apply projections before and after the RMS-norm to reduce the model.

We group the eigenvectors $\boldsymbol{f}$ in percentiles according to the magnitude of their eigenvalues and project the signal prior to the norm into a lower dimensional space using a projection matrix $P$, which contains all eigenvectors $\boldsymbol{f}$ except for the excluded percentile as columns. After the layer norm, $P^T$ is used to rotate back into the original space to keep the network compatible with the following matrices.

Reducing BERT in rank groups of eigenvectors, leads to similar behavior as for the rank group reduction of singular values in the main text, as shown in Fig. 14. When reducing first, we find no relevance in the smaller eigenvectors. However, when fine-tuning first, we see that the smallest eigenvectors are more important than three of the larger percentiles, showcasing their relevance.

