# OpenReview forum: "Locating Information in Large Language Models via Random Matrix Theory"
_ICLR.cc/2025/Conference — Submitted to ICLR 2025_

### Official Review · Reviewer_ZVcU · 2024-10-29

**Soundness:** 3
**Presentation:** 3
**Contribution:** 3
**Rating:** 6
**Confidence:** 4

**Summary:**

This paper presents a novel application of Random Matrix Theory (RMT) to analyze and understand the internal structure of large language models like BERT and Llama-8B. The authors use RMT as a theoretical baseline to identify regions where weight matrices deviate from random initialization, indicating areas of learned features. They validate their findings by comparing with activation covariance matrices and through ablation studies. The work also provides valuable insights into the ongoing debate about the importance of small singular values in fine-tuned models.

**Strengths:**

- The paper is well-organized and provides comprehensive experimental evidence for its claims.

- It offers concrete insights about which matrix types predominantly engage in feature learning and presents an intriguing perspective on the role of small singular values in fine-tuning.

- The paper introduces a novel use of RMT to dissect and understand the internal mechanics of LLMs like BERT and Llama-8B, providing fresh insights into the models' inner structures.

- The study identifies consistent deviations from RMT predictions across different layers and architectures, highlighting a potentially universal structural pattern in language models.

**Weaknesses:**

1. **Lack of Theoretical Derivations:**

   - The authors observe that query and key matrices exhibit pronounced deviations from RMT, indicating feature learning, whereas attention.output and value matrices do not. However, the underlying mechanisms driving these deviations remain unexplored.

   - The discussion around the importance of small singular values post-fine-tuning is intriguing. However, the paper currently relies on empirical results to support this claim. Can the author provide more explanation?

2. **Insufficient Explanation of Methodological Choices:**

    The relationship between p-values derived from Kolmogorov-Smirnov (KS) tests and the model's performance isn't thoroughly explained in Section 4. While lower p-values indicate that the singular vectors deviate from the expected distribution under the null hypothesis, the practical implications of these deviations on learning capacity and generalization remain unclear.

**Questions:**

**[Q1]:** What are the underlying mechanisms driving the observation that specific weight matrices, such as the query and key matrices, engage more in feature learning and deviate more from RMT predictions compared to others like the attention.output and value matrices? Can the authors discuss this in more depth?

**[Q2]:** In Section 4, the paper employs p-values derived from Kolmogorov-Smirnov (KS) tests to assess the randomness of singular vectors' entries. However, how do these p-values directly correlate with the model's performance metrics? Are there empirical correlations between p-values and changes in validation accuracy or loss?

**[Q3]:** Is there a theoretical relationship between the degree of RMT deviation and the model's capacity to learn and generalize? How might the extent of deviation from RMT predictions influence the overall performance and capability of the model?

---

> ### Author Response · Authors · 2024-11-24
>
> We thank the reviewer for their positive and encouraging feedback, and for their time and effort in evaluating our manuscript.
>
> 1) The underlying mechanism why certain matrices engage more in feature learning than others is not shown in our paper, but we speculate that learning the relations between words, as done by the query matrix, may be a demanding task compared to encoding the already embedded token (which is the role of the value matrix).
>
> 2) The correlations between p-values and a decrease in the vallidation accuracy when removing certain groups of singular values that correspond to the p-values is displayed in Figure 7. We show the change in vallidation accuracy and find that in the case of the intermediate dense matrix, where the singular vectors that correspond to the smallest singular values deviate strongly from RMT have a significant effect on the model preformance.
>
> 3) This is an interesting question, but unfortunately there are only empirical results on this. In principle a network which is extremely wide could approximate functions with extremely small changes in its own weights. This is further elaborated in [1]. Reference [1] also states that empirically, lazy learning is not desirable from a generalization standpoint.
>
>
>
> [1] Chizat et all 2019 On lazy training in differentiable programming

---

### Official Review · Reviewer_SF9V · 2024-11-01

**Soundness:** 3
**Presentation:** 3
**Contribution:** 2
**Rating:** 5
**Confidence:** 4

**Summary:**

In this paper, the authors analyze the transformer blocks of a pretrained model (in this case, BERT and Llama) in order to identify the regions of the model where feature learning predominantly occurs. They do so by measuring the discrepancies between quantified aspects of the trained weights in comparison to the expected qualities predicted utilizing RMT methods. The authors claim that regions of the model where the weights significantly differ from RMT predictions correspond to regions where the most learning is occurring.

First, the authors analyze the spectra of the model weights themselves, in particular comparing the distribution of singular values of the trained weights to the distribution predicted by the MP Law. The observation is that in both models, the true distribution differs significantly from the MP distribution.

They then conduct Kolmogorov-Smirno tests on the singular vectors of the weight matrices, quantifying the discrepancies of the distribution of singular vector inputs which are known to follow a normal distribution. The headlining observation of this section is the large deviations from the predicted distribution for both large singular values (not surprising), but also for the smaller singular values, emphasizing the importance of small singular values. This is observed in both models.

The authors then go beyond analyzing intrinsic properties of the weights and explore the alignment between input data to a given layer and the weight matrices themselves. By performing an eigen-analysis of the hidden representations of input data, they quantify the similarity between these eigenvectors and the singular vectors of the weight matrices.

Utilizing the above insights, the authors conclude by analyzing the effects on model performance where removing various decile groups. In all cases the observation is that removing large singular values for any of the weight matrices results in large performance reductions. In most cases, removing the smallest decile of singular values results in negligible performance reductions, apart from the intermediate.dense matrices. Even in this case, however, the reduction is still small compared to the performance hit taken from removing the largest decile. Moreover, the authors compare the effect of fine-tuning both before and after the singular value reduction. In the latter case, strong performance is maintained when removing small singular values. In the former case, however, large performance reductions are observed when removing only the smallest decile.

**Strengths:**

- The paper is well written. It’s easy to follow the equations and the ideas presented.
- The measurements in Section 4 are novel as far as I’m aware. I haven’t seen previous work measuring the normality of the singular vectors.
- The measurements in Section 6 are also novel as far as I’m aware. I haven’t seen previous work measuring the effect of singular vector/value amputation (perhaps with the exception of the pruning literature).
- The measurements in Section 7 are interesting and I haven’t previously seen a similar experiment although it could exist in the literature.

**Weaknesses:**

- The authors state “Following up on this work, Martin et al. (2021) suggested that large outliers in the singular value spectrum are indicative of well-trained matrices.” These outliers were in fact already previously studied in detail in several works. See the works cited in “Traces of Class/Cross-Class Structure Pervade Deep Learning Spectra” and ideally a more recent work on the topic.
- The observation in Figure 7 — that the outlier singular vectors are most important for performance — is explained by the works mentioned in the previous bullet point. Namely, the outliers correspond to class means, which are obviously needed for a well performing model.
- Section 3 essentially extends the measurements of “Implicit Self-Regularization in Deep Neural Networks: Evidence from Random Matrix Theory and Implications for Learning” to LLMs.
- Only the BERT model is analyzed in Sections 6 and 7. This section offers the bulk of the contributions but fails to analyze the Llama model entirely. Figure 9 in the appendix displays similar experiments for Llama, but under very different circumstances. Which SV are being removed here? The lowest percentage? Random removal? Moreover, nowhere are the architectural differences between the BERT and Llama models mentioned. Does the causal nature of masked LLMs play a significant role in your results?
- The authors present results for either hand-picked layers (e.g. Figures 1 and 5), or averages over all layers (Figures 2, 6, etc). As stated in the related work section (line 91), many previous works have revealed significant discrepancies across layers. This is not taken into consideration when undergoing the experiments of Sections 5, 6, and 7. This could have major impacts on the pruning results and overlooks the possibility that some layers could be acting as outliers to the rest of the network.
- Section 5: There is little explanation as to why we would expect there to be any alignment between the input representations $x^l$ and the singular vectors of the weight matrices $W^l$ themselves, and why this would correspond to a learned weight. My intuition would say that it would be more relevant to look at the outputs $F_{nm}^(l+1)$ and compare those eigenvectors to W^l. But then there is also the question of the contribution of the skip connection.
- Is Equation 8 the optimal metric?
    - Goal: directions occupied by the hidden representations should be aligned with the singular vectors of the weights themselves.
    - Since this method takes only the maximum value for each vector, it overlooks the possibility that a vector might be equally aligned with multiple directions. For example, just rotate the standard basis in R^2 by pi/4 radians and compare those two sets of vectors using this measurement.
    - Alternatively, you could look at the singular values of $(F^l)^T  W^l$, which represent the cosine of the principal angles between the subspaces.
- The more general claim is the small singular values carry strong importance, but very little performance is sacrificed when removing the lowest singular values of most of the weights. Even in the intermediate.dense matrix, removing the large SV carries much more weight than removing the small SV (noting the discrepancy in y-axis scale on this plot compared to the others).
- Figure 6. The claim is that large differences of SV entry distributions from the RMT prediction correlate to max eigenvector overlap and suggest feature learning. But all distributions in the upper column differ fairly significantly in all weights, particularly in the output.dense matrix which isnt discussed.
- Figure 7: I doubt that 0 performance was lost when removing most of the ranked groups. Perhaps a log scale on the y-axis would produce a higher fidelity plot. Also, are the inset plots not analogous to Figure 4? I think it would be good to mention this.
- Figure 8: What do the horizontal dashed lines represent?
- Equation 4: This is the distribution of the magnitude of the entries of the singular values, right?

**Questions:**

- “…a few outliers and are dominated by regularization” — I don’t understand what does this mean.
- “the query matrices display significant outliers for both the Llama-8B and BERT” — I don’t see outliers in the query matrix of Llama.
- “We interpret this behavior as an indication that feature learning predominantly occurs in the query matrices, where the weights undergo substantial changes, while the attention.output matrices remain closer to their initial random state, reflecting lazy learning.” — this seems rather strange. why would that be the case? Do you an explanation for this observation or at least a hypothesis?
- “intermediate.dense matrix” — what does this mean?
- For multi head attention, are measurements done for the weight matrices associated with a single head or the weight matrix obtained by concatenating all heads? If concatenation, could it be that attention.output looks random simply because of the fact that you’re concatenating all heads?
- For the intermediate.dense matrix, why would the top singular vectors appear more random than the middle singular vectors? Could it be related to the fact that you’re concatenating all heads into a single matrix?
- “BoolQ training dataset” mentioned without a citation. Also, why specifically this dataset instead of a more general purpose dataset?
- In Figure 5 (b), could the authors add the vertical black dashed line (similar to Figure 4)?
- Figures 3 and 4 show that many singular vectors (far more than just those corresponding to the outliers) deviate from normality. Figure 5 and 6 show that only the top and bottom singular vectors of the weight matrices are actually coupled with the features. These statements seem to contradict one another. How does one reconcile these observations?
- Related to the previous point, the following statement in conclusion seems inaccurate: “a strong overlap between the weight’s singular vectors in regions that deviate from RMT predictions and the eigenvectors of the corresponding covariance matrix of activations entering the layer.”.
- The authors keep mentioning “lazy regime” and “feature learning regime”. Could you provide a definition for them?
- “These deviations from RMT are consistent across all layers” — how do we know that if measurements were only presented for a very small subset of weight matrices?
- Figure 3a: Does this not say that the trained SV are precisely as predicted for a randomly initialized matrix?
- Figure 2: This is analogous to Figure 1 with the average over all blocks being plotted instead. A common theme in this paper seems to be treating each block as equal. I would be very interested in the evolution of these metrics as a function of depth. Also figure 1 doesn’t offer much more information than Figure 2. Including both of these figures seems a bit redundant, especially considering the significance of some of the plots in the appendix (which I discuss bellow).
- A common term when referring to large singular values is “outlier”. Is this the proper term, considering the singular values appear to follow a fairly smooth distribution in almost all cases?
- Figure 5 is for the first block only? I would be much more interested in the average across layers, or in particular, not the first layer since it is not particularly representative of the network as a whole, especially for the attention matrices.
- Figure 4: query and intermediate.dense matrices show large correlation with the untrained variants, while the attention matrix shows the overall largest deviations. Does this exactly support the claims that feature learning occurs in query and intermediate, and not in attention?
- You briefly mention training dynamics in the related work section. It would be interesting to observe how your metrics, particularly deviations from MP predictions, evolve throughout training by using, for example, the Pythia checkpoints.
- Figure 6 (bottom row): It is interesting that all weight matrices plateau at the same constant value, except for output.dense
- Figure 6. You are comparing the input activations to a given layer and the weight matrices of that layer. But by the time you are comparing against the output matrix, the representations have already flowed through the transformer block. Would it not make sense to compute alignment of the output of the attention block and the singular vectors of the output matrix?
- Figure 8: Very interesting observation, I like this figure.
    - Reduce then tune: This intuitively makes sense to me; fine-tuning preserves the overall structure.
    - Tune then reduce: This also makes sense to me, but why are we missing the plot points for removing larger SV?
    - In all cases, are you removing the i^th decile from all weight matrices in the entire network?
    - It would be interesting to “hack” the best possible combination of singular values to remove. For example, you could try removing the lowest singular values for all matrices except the intermediate.dense ones, where you only remove the middle-magnitude SVs. This is backed by the results of Figure 7.
    - If you re-fine-tune the models displayed on the right figure after removing the smallest SV, would you retain the original performance shown on the left?
- Stylistic choice: I prefer a legend on my plots, rather than having to dig through the figure captions.

---

> ### Author Response · Authors · 2024-11-24
>
> We thank the Reviewer for their positive feedback.
> We further address the Reviewers points and suggestions below.
>
> 1) We apologize for the misleading formulation. We meant to say that in the case of Llama-8B, there is strong regularization used during training which shifts the singular values to smaller values. We have adjusted the text in the manuscript.
>
> 2) We speculate that learning the relations between words, as done by the query matrix, may be a demanding task compared to encoding the already embedded token (value matrix).
>
> 3) We apologize for not specifying precisely where in the architecture of BERT the "intermediate.dense" matrix resides. It is the first dense matrix after each attention block, following the attention.output.
>
> 4) We concatenate the heads to get a large enough matrix to draw statistically significant conclusions. We find significant deviations in the query matrix compared to very small deviations in the value matrix, showcasing that the concatenation does not hinder our analysis.
>
> 5) The intermediate.dense matrix is not part of the attention mechanism but of the feed-forward neural network after the attention block. In this case, no heads are concatenated.
>
> 6) We thank the reviewer for their remark and added a  citation for the BoolQ dataset. We use BoolQ as binary question-answering tasks aligns well with our study's goal of evaluating how transformers encode and utilize information. However, we acknowledge that using a more general-purpose dataset would strengthen the generality of our findings.
>
>
> 7) Yes, we added the vertical dashed line to an updated version of Figure 5 (b).
>
> 8) Figure 3 and 4 indeed show that singular values deviate from the baseline in both directions, meaning they often have increased or lowered p-values. However, we only consider a decreased p-value as a sign of learned features. The reason for increased p-values is a small mean in the singular vectors corresponding to features. As the singular vectors are orthogonal to each other, other singular vectors have a mean of precisely zero, leading to increased p-values.
> In this picture, the results are in good agreement with Figure 5 and 6.
>
> 9) A definition of lazy and feature learning is found in Appendix A. We tried to make this more visible in the updated manuscript.
>
> 10) While we do not show each individual layer we present averages over all layers in Figure 2, 4, and 6. These averages agree well with the results of individual matrices demonstrating that the shown examples are typical for a given layer.
>
> 11) The reviewer is correct that Figure 3a shows an example of a singular vector that is very close to the expected distribution and hence has a high p-value.
>
> 12) We thank the reviewer for pointing out their interest in the evolution of the weight matrices with layer depth. We found it rather interesting that our analysis is so consistent across all layers. However, we will add a plot as a function of the  layer index to the appendix to show that this is indeed the case.
>
> 13) We refer to singular values as "outliers" when they are larger than the MP-boundary and added this definition to the updated manuscript.
>
> 14) Figure 5 is indeed showing the first block, however, the average that the question aims at is shown in Figure 6.
>
> 15) Figure 4 shows the average p-value over all blocks (orange line) and the p-values from the fourth block (blue line). We find that in the case of the query and attention.output matrix, there are deviations only for the largest singular values, while the intermediate dense matrix has deviations for the smallest singular values. Could the reviewer elaborate in what sense the attention.output matrix has the largest deviations?
>
> 16) We thank the reviewer for pointing out this interesting option an will add a discussion of the training dynamics in a future version.
>
> 17) This is due to the fact that the dimension in which the scalar product is computed is larger in the case of the output.dense matrix.
>
> 18) "Would it not make sense to compute alignment of the output of the attention block and the singular vectors of the output matrix?" This is precisely what we are doing. For each matrix type, we use the input activations that the matrix receives to compute the activation covariance matrix and compare to the singular vectors of this matrix.
>
> 19) Figure 8: Yes, we always remove the i-th decile from all weight matrices in the entire network. In the plot, the missing points are outisde of the y-limits, meaning that for the largest singular values the performance decreses strongly. We added a note about this to the manuscript. We also thank the reviewer for the suggestions, which we will add in a future version.
>
> We want to thank the reviewer and believe that the suggestions will help us refine our work and strengthen its impact.

---

> > ### Comment · Reviewer_SF9V · 2024-11-26
> >
> > The authors have acknowledged the concerns raised; however, they have not addressed any of the identified weaknesses. Furthermore, as far as I can determine, the manuscript has not been updated, despite the opportunity to make improvements.

---

### Official Review · Reviewer_xWdk · 2024-11-04

**Soundness:** 2
**Presentation:** 3
**Contribution:** 2
**Rating:** 5
**Confidence:** 3

**Summary:**

This paper uses spectral analysis to probe the role that small singular values have in weight matrices of language models. They use ideas from random matrix theory to identify singular components that are similar to randomly initialized matrices, and take these as an indicator of when and where feature learning occurs. They use this information to study the consequences of removing small singular vectors, which has come up in recent literature on the subject.

**Strengths:**

The idea of using RMT is much more efficient than what I've seen in previous work. In the LASER [1] paper, for example, the authors do a search over how the removal of singular components affects performance, which can be expensive.

1. Sharma, P., Ash, J. T., & Misra, D. (2023). The truth is in there: Improving reasoning in language models with layer-selective rank reduction. ICLR 2024.

**Weaknesses:**

The analysis performed here is quite limited, and is shown for only two dataset-model pairs. As such, I'm not certain whether the results described here are robust to more general cases in which LLMs are used. Another concern I have is in the section where singular vectors are removed from weight matrices. The authors say they "removed one of the singular value deciles from a specific matrix type in all layers,” but I don't believe this is standard. In [1], for example, the intervention is applied only at a specific matrix, leaving the singular vectors most of the weight matrices unchanged. I'm not sure whether the coarse intervention applied here actually provides insight into the utility of small singular values, because this seems to be a large deviation with previous work.

**Questions:**

1. Do these results hold for more model-dataset pairs? Is it a general phenomenon?
2. If we instead intervene at only one matrix, leaving all others the same as is done in previous work, do the conclusions hold?
3. How do the spectral properties discussed here change as model training progresses?

---

> ### Author Response · Authors · 2024-11-24
>
> We thank the reviewer for their thoughtful feedback. We appreciate the time to assess our work and highlight its strengths, including the efficiency of our method.
> We answer the questions raised by the reviewer below:
>
> 1) We believe that the observed phenomena are of generic nature as both a BERT transformer and a Llama-8B model have similar patterns regarding their weight matrices. We plan to add a evaluation of the Llama-8B model on more datasets in a future version.
>
> 2) In Figure 2 of "The truth is in there: Improving reasoning in language models with layer-selective rank reduction. ICLR 2024" the authors remove significant portions (95%, 99%, 99.5%) of individual matrices. Even when removing 50% of the singular values in a specific matrix there were only insignificant changes in the loss. To have a significant signal we therefore needed to remove singular values at minimum from all matrices that share a common type, e.g. all query matrices.
>
> 3) We thank the reviewer for their interest in how spectral properties change during training. We did not analyze this problem yet, but we will add it to the discussion in a future version.

---

### Official Review · Reviewer_szca · 2024-11-04

**Soundness:** 2
**Presentation:** 2
**Contribution:** 2
**Rating:** 5
**Confidence:** 3

**Summary:**

This paper uses random matrix theory (RMT) to analyze the weight matrices of BERT and Llama-8B models. The key technical contributions are as follows:

1. A method has been developed to identify learned features in transformers by detecting deviations from the Marchenko-Pastur distribution in singular value spectra and from Gaussian distributions in singular vectors.
2. It is demonstrated that query matrices exhibit significant deviations from RMT predictions, while attention matrices remain close to random. This pattern is consistent across different architectures.
3. The method is validated by showing a high overlap between regions that deviate from RMT and the eigenvectors of the activation covariance matrix.
4. Evidence is provided that small singular values, although not essential during pre-training, become critical during fine-tuning, as shown through systematic ablation studies.

________
Update: After reviewing the responses, I hold my original scores.

**Strengths:**

**Originality:**
- Novel use of random matrix theory as a zero-information baseline to identify learned features in transformers, revealing consistent patterns across architectures (query vs attention.output matrices)
- Development of a systematic methodology combining singular value analysis with activation covariance validation, providing a way to study model internals without requiring training data

**Quality:**
- Rigorous empirical validation through dual verification: first identifying deviations from Marchenko-Pastur distribution, then confirming these regions have high overlap with eigenvectors of activation covariance matrices
- Comprehensive ablation studies across different matrix types and model scales (BERT to Llama-8B), with proper statistical controls and error analysis over multiple fine-tuning runs

**Clarity:**
- Well-structured progression from theoretical foundations to empirical validation, with each section building logically on previous findings
- Clear visualization strategy, particularly in Figures 5-6, showing both the singular value spectra and their correspondence to learned features through activation covariance analysis

**Significance:**
- Resolution of the small singular values debate by demonstrating their importance emerges during fine-tuning, with concrete implications for model compression strategies
- Development of a diagnostic tool for identifying learned features using only weight matrices, enabling analysis of black-box models without access to training data or activations

**Weaknesses:**

# Technical Limitations

1. **Non-Linear Component Limitations**
If I understand correctly, the analysis theory is fundamentally limited by focusing only on linear weight matrices $W = USV^T$. The paper fails to address how non-linearities (GELUs, LayerNorms) affect and potentially invalidate RMT predictions. This is particularly problematic as the MP distribution assumptions strictly apply only to linear operations, and the interaction between non-linear activations and the singular value spectrum remains uncharacterized.

2. **Statistical Weaknesses in Activation Analysis**
The validation methodology using activation covariance matrices suffers from significant statistical limitations. The analysis relies solely on the BoolQ dataset without establishing statistical significance measures, confidence intervals, or minimum sample size requirements. There's no rigorous treatment of sampling noise affecting the observed correlations between activation covariance eigenvectors and singular vectors.

3. **Scale and Architecture Dependencies**
The paper does not systematically investigate how RMT deviations depend on fundamental parameters. While results from BERT and Llama-8B are presented, scaling behaviors with respect to model size, matrix dimensions, or training corpus size are not analyzed. This absence makes it impossible to determine whether the observed patterns are universal or specific to the studied architectures and scales.

4. **Fine-Tuning Dynamics Gap**
Analyzing fine-tuning effects is limited to static comparisons of pre- and post-fine-tuning states. The paper doesn't investigate the dynamical evolution of singular values and their corresponding vectors during fine-tuning. Without this temporal analysis, the mechanism by which small singular values become important during fine-tuning remains unclear, limiting the theoretical understanding of the phenomenon.

5. **Theoretical Bounds Absence**
The work lacks theoretical bounds on expected deviations from RMT predictions. While empirical deviations are documented, there's no formal framework for determining when these deviations become statistically significant. The absence of concentration inequalities for singular value distributions and finite-size corrections limits the rigor of the statistical analysis, making it difficult to establish confidence in the observed patterns.

**Questions:**

Thanks for the submission; I have some questions and suggestions for the authors

1. **Theoretical Justification of Non-Linear Effects**
Could you clarify how the RMT predictions remain valid when analyzing weight matrices embedded in non-linear architectures? Specifically, how do LayerNorm and GELU operations affect the theoretical expectations from the Marchenko-Pastur distribution? A mathematical treatment showing why the linear analysis remains informative would strengthen the theoretical foundations.

2. **Statistical Significance Framework**
What statistical framework would you propose to establish the significance of RMT deviations? It would be valuable to see:
- The minimum deviation threshold is considered meaningful
- How you account for finite-size effects in your analysis
- Whether the observed patterns in activation covariance overlaps remain consistent across different random seeds and datasets

3. **Fine-Tuning Evolution**
Could you provide data on how singular values and their corresponding vectors evolve during fine-tuning? This would help understand:
- The trajectory of initially small singular values that become important
- Whether the changes are gradual or exhibit phase transitions
- If there's a correlation between the magnitude of RMT deviation and the impact on downstream task performance

4. **Scaling Behavior Clarification**
While you show results for BERT and Llama-8B, could you clarify whether the magnitude of RMT deviations scales systematically with model size? Specifically, does the relative importance of different matrix types (query vs attention.output) remain constant across scales, and do you observe any consistent patterns in how deviations from MP distribution scale with matrix dimensions?

5. **Validation Methodology**
- What alternative validation methods did you consider beyond activation covariance analysis?
- The current validation relies heavily on BoolQ dataset correlations. Could you discuss whether techniques like probing or intervention studies might provide complementary evidence for your RMT-based feature identification?

---

> ### Author Response · Authors · 2024-11-24
>
> We thank the Reviewer for their thoughtful and detailed feedback. We appreciate the effort to assess our work and highlight its strengths, including the originality of our methodology, the rigor of our empirical validation, and the clarity of our presentation.
> We answer the questions raised by the Reviewer below:
>
> 1) Non-linear effects: The effect of non-linearities are included in our numerical analysis is studied in the activation-covariance matrices, which are computed before these activations enter a linear layer again (hence directly after the nonlinearity). We find that the neuron activations after this nonlinearity have a significant overlap with the linear operation, i.e. the weight matrix.
>
>
> 2) We agree with the Reviewer that finding a threshold that we consider as statistically significant would benefit the work. Further, we believe that the overlap between the weights and the activation covariance matrix computed on larger amounts of text is generic and does not depend on the specifc text at hand, however, we will add additional plots to verify this in a future version.
> In addition, we show in Figure 1 that the weight matrices are large enought to have neglectible finite size effects.
>
> 3) We thank the reviewer for pointing out this interesting research idea. Analyzing the dynamics of the RMT-properties during training could certainly add to the discussion and we will add this to a future version.
>
>
> 4) Unfortunately, our current setup  cannot conclusively answer this interesting question as two datapoints with different model sizes do not allow for a scaling analyzis. Particularly if the architectures are that different.
>
>
> 5) We agree that further evidence with more datasets would be beneficial. We provide a small intervention study when considering the effect of the removal of smaller singular values. However, probing could additionaly be used to test whether the regions of the weight matrix identified as "deviating" from RMT correspond to representations that encode specific linguistic or task-related features.
>
>
> We thank the Reviewer and we believe the suggestions will help us refine our work and strengthen its impact.

---

### Official Review · Reviewer_mws8 · 2024-11-08

**Soundness:** 3
**Presentation:** 2
**Contribution:** 1
**Rating:** 3
**Confidence:** 4

**Summary:**

In this paper the authors explore the eigenspectra of weight matrices and activations in transformer models and how they deviate from the RMT derived behavior from their random initialization. Analyzing the eigenvalues and eigenvectors in different ways, the authors identify statistically significant deviations at the top and bottom of the spectrum and they use these results to make statements about feature learning in these networks.

**Strengths:**

The writing in large part is adequate with no major issues that I noticed. The experiments are fairly clear, along with presentation. I appreciate the effort that the authors have invested to make their experiments reproducible. The related work section is mostly adequate, though I believe that Greg Yang's Tensor Programs and MuP in the feature learning limit should be mentioned and cited, and probably discussion of the NTK would be warranted also.

**Weaknesses:**

I did not feel that the investigation undertaken in this paper provided substantial insights into transformers/LLMs. Much of the analysis is ostensibly at the level of analysis of: we show that statistics of weight and activation matrices differs from their random initializations. I don't think this point is in contention. The fact that the eigenvectors differ most from random gaussians top and bottom of the spectrum adds to this slightly, but I did not see an effort to explain this mathematically. In general the tools of RMT appear to be applied only in a somewhat superficial manner: as an analytic distribution to be used as the null hypothesis (which could even have been done just through sampling without the analytic form). From this kind of paper I would have expected to see RMT tools levered to better explain what does actually happen in the transformer weight matrices. The empirically measured impact of removing singular values seems to have little to do with the RMT analysis, and for a paper titled as locating information in LLMs I would expect information content to be measured and quantified. How is information spread across layers, how is information distributed across the spectrum of the matrices, how does it change across the course of training, these are the kinds of questions that I would expect to be addressed in a quantitative manner.

The largest drawback of the paper in my opinion is the lacking in significance and extent of the scientific contributions uncovered here.

**Questions:**

105 "while transformer features often exhibit low rank" (grammar?)


182: "the attention.output matrices" (presumably unintentional? this attention.output is repeated many times and should not be formatted in this way)
also line 184, figure 3 caption


equation 7: shouldn't the covariance matrix be centered (mean subtracted)?

This does not impact my evaluation, but a word of advice to the authors: I would always avoid using the default matplotlib blue orange colors. This choice serves as an unconscious signal that the figures are low quality or that the authors were not very intentional about the way the information is presented even if this is not the case.

---

> ### Author Response · Authors · 2024-11-24
>
> We thank the Reviewer for their detailed feedback and for acknowledging the clarity of our experiments, reproducibility efforts, and presentation quality. We also appreciate the suggestions for improving the related work section and will ensure to incorporate the mentioned work in our literature discussion.
>
> We agree that providing deeper insight into "what does actually happen in the transformer weight matrices" is the ultimate question. We contribute to answering this question by showing that a certain group of matrices is not actively participating in feature learning, namely the value matrix and the dense matrix following the attention mechanism (named attention.output in our notation). This is demonstrated by showing that the singular values of these weight matrices do not exceed the theoretical Marchenko-Pastur boundary, and additionally the singular vectors have very little overlap with the activation-covariance matrix eigenvectors (Figure 6).
>
> We are sorry that the notation "attention.output" is irritating. The notation comes from a programming side and is the official naming of the weights in the huggingface pytoch model.
>
>
> We changed "while transformer features often exhibit low rank"  to "the feature matrices of transformers are often low rank".
>
>
> Yes the reviewer is absolutely right, the covariance matrix should be centered around 0. We thank the reviewer for their feedback and believe that implementing the reviewers suggestions will help us refine our work and strengthen its impact.

---

### Meta-Review · Area_Chair_U6CN · 2024-12-21

**Metareview:**

### Summary
This paper examines the weight matrices of pre-trained transformer models, specifically BERT and Llama, using random matrix theory (RMT) as a baseline for analyzing structural deviations introduced during training. The authors find that while weights in randomly initialized models align with RMT predictions, trained models exhibit deviations that highlight areas of significant feature learning. These deviations are consistent across different layer types and architectures. The analysis focuses on the singular value distributions of weight matrices, showing notable differences from RMT predictions, especially for small singular values. The authors demonstrate that these small singular values are crucial for model performance, particularly after fine-tuning, and removing them can negatively impact alignment and capabilities. Additionally, the study explores the relationship between the learned weights and activation patterns, linking the observed deviations to areas of meaningful feature extraction. The findings suggest that small singular values contribute more to performance than previously understood and caution against their removal in fine-tuned models. This work introduces a diagnostic approach for interpreting transformer models' structure and learning dynamics using trained weight matrices alone.

### Decision

The paper is being rejected due to several weaknesses identified by the reviewers. These weaknesses are grouped below based on common themes for clarity.

1. Lack of Depth in Analysis

	•	Reviewer mws8 highlighted that using random matrix theory (RMT) was superficial and limited to serving as a null hypothesis. The paper failed to leverage RMT to provide deeper mathematical insights or explain the observed phenomena in transformer weights.

	•	Reviewer SF9V pointed out that the paper broadly extends prior work on RMT in deep learning without adding significant new contributions, particularly in analyzing large language models (LLMs). Key architectural differences between BERT and Llama were not adequately addressed.

2. Insufficient Consideration of Non-linearities

	•	Reviewer szca noted that the analysis is fundamentally limited by focusing only on linear weight matrices, without addressing how non-linearities (such as GELUs and LayerNorms) affect the RMT predictions. This omission weakens the validity of the analysis.

3. Statistical and Methodological Weaknesses

	•	Reviewer szca criticized the lack of rigorous statistical validation, including the absence of confidence intervals, statistical significance measures, and an analysis of sampling noise. The paper relied solely on one dataset (BoolQ), limiting the generalizability of its findings.

	•	Reviewer xWdk expressed concerns about the coarse intervention method for removing singular vectors, which deviates from established practices in prior work, raising doubts about the validity of the results.

4. Limited Scope and Generalizability

	•	Reviewer xWdk observed that the experiments were conducted on only two dataset-model pairs, making it unclear whether the results are robust across different models and tasks.

	•	Reviewer szca mentioned that the paper does not explore how RMT deviations vary with model size, architecture, or training corpus, leaving open questions about the universality of the findings.

5. Inadequate Exploration of Fine-tuning Dynamics

	•	Reviewer szca noted that the analysis of fine-tuning effects was limited to static comparisons before and after fine-tuning, without investigating the evolution of singular values during fine-tuning. This gap limits the theoretical understanding of why small singular values become important.

6. Ambiguities in Experimental Setup

	•	Reviewer SF9V raised questions about the pruning experiments, including whether the removal of singular values was applied randomly or selectively. Additionally, layer-specific differences were not considered, which could affect the results.

	•	Reviewer xWdk pointed out that the explanation of why input representations should align with singular vectors of the weights was unclear. The contribution of skip connections was also overlooked.

7. Presentation Issues

	•	Reviewer SF9V identified unclear explanations for key results, including ambiguities in the figures and metrics. For example, the alignment metric in Equation 8 may not capture the phenomenon accurately, and some figures lack adequate labeling or explanation.

#### Conclusion

The reviewers agree that the paper lacks depth, rigor, and scope in its analysis. While using RMT in analyzing LLMs is a promising direction, this work does not provide sufficient scientific contributions or robust experimental evidence to justify acceptance. These limitations significantly reduce the impact and clarity of the work.

**Additional Comments On Reviewer Discussion:**

The reviewers, in general, agreed that the topic the paper studies is a promising and interesting paper. However, it feels rushed and, as it stands, lacks scientific rigor and is not ready for publication. They pointed out several weaknesses of the paper. However, the rebuttal submitted by the authors was somewhat underwhelming. Thus, the reviewers agreed that this paper is below the acceptance bar.

---

### Decision · Program_Chairs · 2025-01-22

Reject